# In Silico Characterisation of Putative Prophages in *Lactobacillaceae* Used in Probiotics for Vaginal Health

**DOI:** 10.3390/microorganisms10020214

**Published:** 2022-01-20

**Authors:** Anna-Ursula Happel, Brian R. Kullin, Hoyam Gamieldien, Heather B. Jaspan, Arvind Varsani, Darren Martin, Jo-Ann S. Passmore, Rémy Froissart

**Affiliations:** 1Department of Pathology, Institute of Infectious Diseases and Molecular Medicine, University of Cape Town, Anzio Road, Cape Town 7925, South Africa; anna.happel@uct.ac.za (A.-U.H.); brian.kullin@uct.ac.za (B.R.K.); hoyam.gamieldien@uct.ac.za (H.G.); hbjaspan@gmail.com (H.B.J.); jo-ann.passmore@uct.ac.za (J.-A.S.P.); 2Seattle Children’s Research Institute, 307 Westlake Ave. N, Seattle, WA 98109, USA; 3Department of Pediatrics and Global Health, University of Washington, 1410 NE Campus Parkway NE, Seattle, WA 98195, USA; 4The Biodesign Center of Fundamental and Applied Microbiomics, Center for Evolution and Medicine, School of Life Sciences, Arizona State University, 1001 S. McAllister Ave., Tempe, AZ 85281, USA; arvind.varsani@asu.edu; 5Structural Biology Research Unit, Department of Integrative Biomedical Sciences, Institute of Infectious Diseases and Molecular Medicine, University of Cape Town, Anzio Road, Cape Town 7925, South Africa; 6Division of Computational Biology, Department of Integrative Biomedical Sciences, Institute of Infectious Diseases and Molecular Medicine, University of Cape Town, Anzio Road, Cape Town 7925, South Africa; darrenpatrickmartin@gmail.com; 7NRF-DST CAPRISA Centre of Excellence in HIV Prevention, 719 Umbilo Road, Congella, Durban 4013, South Africa; 8National Health Laboratory Service, Cape Town 7925, South Africa; 9CNRS, IRD, Université Montpellier, UMR 5290, MIVEGEC, 34394 Montpellier, France

**Keywords:** genital, prophages, bacteriophages, probiotics, antibiotic resistant genes, virulence factor

## Abstract

While live biotherapeutics offer a promising approach to optimizing vaginal microbiota, the presence of functional prophages within introduced *Lactobacillaceae* strains could impact their safety and efficacy. We evaluated the presence of prophages in 895 publicly available *Lactobacillaceae* genomes using Phaster, Phigaro, Phispy, Prophet and Virsorter. Prophages were identified according to stringent (detected by ≥4 methods) or lenient criteria (detected by ≥2 methods), both with >80% reciprocal sequence overlap. The stringent approach identified 448 prophages within 359 genomes, with 40.1% genomes harbouring at least one prophage, while the lenient approach identified 1671 prophages within 83.7% of the genomes. To confirm our in silico estimates in vitro, we tested for inducible prophages in 57 vaginally-derived and commercial *Lactobacillaceae* isolates and found inducible prophages in 61.4% of the isolates. We characterised the in silico predicted prophages based on weighted gene repertoire relatedness and found that most belonged to the *Siphoviridae* or *Myoviridae* families. ResFam and eggNOG identified four potential antimicrobial resistance genes within the predicted prophages. Our results suggest that while *Lactobacillaceae* prophages seldomly carry clinically concerning genes and thus unlikely a pose a direct risk to human vaginal microbiomes, their high prevalence warrants the characterisation of *Lactobacillaceae* prophages in live biotherapeutics.

## 1. Introduction

Bacterial vaginosis (BV) is a common adverse condition in women of reproductive age [1,2]. It involves a shift from a relatively homogenous *Lactobacillaceae*-dominated cervicovaginal microbiota to one that is polymicrobial and lactic acid bacteria-depleted [3,4]. The standard of care for BV is antibiotics; however, the effective and long-lasting treatment of BV and other clinical dysbioses with antibiotics has proven difficult [5]. Thus, *Lactobacillaceae*-containing probiotics have been explored as an adjunctive therapy [6,7]. *Lactobacillus*, *Lacticaseibacillus*, *Ligilactobacillus, Lactiplantibacillus* and *Limosilactobacillus* spp. are considered to be ideal probiotic candidates for treating BV as they produce antibacterial metabolites [8,9,10], lower vaginal pH [9,11,12], inhibit the adhesion and growth of non-optimal bacteria [12,13,14], and modulate innate immunity [11,12,15].

Probiotic therapies, particularly those targeted at the promotion of women’s reproductive health, have gained increasing interest in the past few years amongst both the scientific community [16] and the general public [17,18]. Whether all *Lactobacillaceae* genera, species, or strains make equally safe and effective probiotics for promoting vaginal health remains a pertinent question.

One factor that could influence the safety and effectiveness of any given *Lactobacillaceae* strain is the presence of functional prophages within their genome [19]. Bacteriophages are the most numerous and diverse organisms on Earth [20] and play important roles both in shaping the structure of bacterial communities and in facilitating the evolution of bacterial genomes [21]. Functionally, bacteriophages are mostly divided into virulent (strict lytic cycle) and temperate (lysogenic cycle) types [22]. In lysogenised bacteria, the prophage can be excised upon exposure to stress or other environmental stimuli, for example, DNA-degrading antibiotics such as quinolones [23], mutagens such as mitomycin C [24,25,26], anticancer, antimicrobial and antiseptic agents [27], or chemicals in cigarette smoke [28], upon which the bacteriophage resumes a lytic cycle [22].

Besides the direct phenotypic consequences of bacteriophage infections, which can include the enhancement of bacterial virulence [29,30,31], bacteriophages share a long evolutionary history with bacteria. Lysogenic bacteriophages have profoundly modified bacterial genomes via their mediation of horizontal gene transfer (HGT) between individual bacteria and have increased the capacity of their hosts to adapt to dynamic environments [32]. Horizontal transfer of antibiotic resistance genes (ARGs) has been reported to occur both in vitro and in vivo in mouse models [33,34,35], humans [36,37,38] but not in pigs [39], and via food consumption [40]. ARGs have also been detected in bacterial strains contained in probiotic supplements [41,42], and probiotics were shown to exacerbate resistome expansion in the gastrointestinal mucosa by supporting the growth of strains carrying vancomycin resistance genes [43]. This raises the question whether bacteriophage-mediated HGT could lead to the transfer of ARGs from lysogenic probiotic strains to resident bacteria [44,45].

Bacteriophages can also disrupt host genes when they insert into host chromosomes, which can lead to alterations in host mutation rates, virulence factors, biofilm regulation, sporulation and phagosome escape [46]. On the contrary, bacterial genotypes harbouring prophages are protected from infections by closely related bacteriophages that belong to the same bacteriophage immunity group [47], thereby potentially providing a growth advantage over bacterial genotypes that lack prophages, which may be advantageous for live biotherapeutic products.

Inducible prophages with varying host ranges and functionalities have been identified using in silico and in vitro approaches in the genomes of various *Lactobacillaceae* genera and species, e.g., in *Lactobacillus gasseri* ADH [48], *Lactobacillus jensenii* Lv-1 [27], *Limosilactobacillus reuteri* [49,50], *Lactiplantibacillus plantarum* [51] and some vaginally-derived strains of *Lactobacillus crispatus*, *Lactobacillus jensenii, Limosilactobacillus fermentum, Lactobacillus acidophilus* and *Lactobacillus gasseri* [24,25,26]. Metagenomic surveys of the human gut [52] and female genital tract (FGT) [53] microbial communities have revealed the presence of known *Lactobacillaceae* bacteriophages, suggesting that they can be induced and actively transmitted in the gut and FGT. Given the various ways in which bacteriophages contribute directly and indirectly to bacterial genomic diversity, it is plausible that they contribute to increased bacterial adaptability in benign [54,55,56] and adverse environments, such as those with high antibiotic or hydrogen peroxide concentrations [56,57]. Thus, the balance between health benefit profiles versus safety impairments needs to be considered in the development of new probiotic candidates.

Previous studies on the lysogenesis of vaginal strains belonging to *Lactobacillaceae* genera and the possibility that prophages may adversely impact the bacterial communities into which they are introduced [12,24,25,26,27,48,49,50,51,58] suggest that an investigation is warranted into both the presence of prophages within probiotic *Lactobacillaceae* candidates, and the genomic contents and functionality of whatever prophages are present. We therefore computationally assessed the presence of prophages within the publicly available genome sequences of 12 species of *Lactobacillaceae* that are commonly present in FGT as well as *Lactobacillaceae* used in probiotics for FGT health. The predicted prophages were assessed in silico for functionality and the presence of known toxins and ARGs. To infer whether these in silico estimates might have any clinical relevance, we determined the prevalence of inducible prophages among *Lactobacillaceae* strains found within current and intended probiotic products for use in the FGT in vitro.

## 2. Materials and Methods

### 2.1. Selection of Publicly Available Lactobacillaceae Genomes from NCBI

Genomes of *Lactobacillaceae* genera commonly present within live therapeutics for vaginal health that were available on 1 June 2019 were downloaded from the NCBI RefSeq assembly database (Appendix A), including genomes of *Lactobacillus acidophilus* (*n* = 37), *Lactobacillus crispatus* (*n* = 91), *Lactobacillus gasseri* (*n* = 29), *Lactobacillus helveticus* (*n* = 53), *Lactobacillus iners* (*n* = 21), *Lactobacillus jensenii* (*n* = 18), *Limosilactobacillus mucosae* (*n* = 11), *Lactiplantibacillus plantarum* (*n* = 434), *Limosilactobacillus reuteri* (*n* = 125), *Lacticaseibacillus rhamnosus* (*n* = 149), *Ligilactobacillus salivarius* (*n* = 82) and *Limosilactobacillus vaginalis* (*n* = 3). This initial dataset (*n* = 1053) was filtered by excluding assemblies with N50 < 10 Kbp (*n* = 9) and >500 ambiguous bases (Ns) per 100 Kbp (*n* = 101). The filtered dataset was assessed using CheckM (version 1.1.3) [59] and assemblies with a completeness score of <90% and/or evidence of contamination at >5% were excluded (*n* = 22). Duplicate sequences defined as sequences originating from the same strain were manually identified and removed (*n* = 24), retaining only the highest quality sequences in each case. This resulted in a final dataset of 897 high-quality genomes.

### 2.2. In Silico Identification of Prophage Sequences in Lactobacillaceae Genomes

Prior to the detection of potential phage sequences, all genome assemblies in the final genome dataset were assayed for the presence of potential PhiX contamination using the blastn option in the NCBI standalone BLAST suite (version 2.2.31) [60]. For detection of prophage sequences, the following algorithms were used: (i) Phaster [61] (default options, limiting hits to bacteriophages classified as ‘Intact’ by the algorithm); (ii) Phigaro (version 0.2.1.5) [62] (default options, using the prokaryotic virus orthologous groups (pVOGs) database [63], removing contigs < 20 Kbp prior to run); (iii) Phispy (version 3.2) [64] (default options, using the generic PhiSpy training set); (iv) Prophet [65] (default options, using the Prophet database (based on proteome sequences of phage genomes available in Genbank)); and (v) Virsorter (version 1.0.5) [66] (default options using the extended Virsorter database (based on sequences present in the RefSeq Virus database supplemented with curated virome sequences), using Diamond (version 2.0.0.138) instead of BLAST for the searches and limiting to ‘confident’ and ‘likely’ hits). Results from each of the algorithms were filtered by removing prophage hits that had >10 Ns in the predicted prophage region and/or those that were not within the size range of bacteriophages in the NCBI *Caudovirales* database (<11,600 bp and >469,500 bp). A consensus approach was then used to generate the final two prophage datasets. In this approach, the results from each of the algorithms were used to prepare BED interval files containing the putative prophage regions for each prophage genome. The ‘lenient’ dataset consisted of prophages identified by ≥2 algorithms and the ‘stringent’ dataset included only prophages that were detected by ≥4 algorithms. Predicted prophage regions in both datasets were further analysed using CheckV (version 0.6.0) [67] to assess viral genome completeness and contamination.

### 2.3. Weighted Gene Repertoire Relatedness

To assign predicted prophages to families, a weighted gene repertoire relatedness (*wGRR*) approach was used, similar to that previously reported [68,69]. The *wGRR* score between each predicted prophage and every bacteriophage in the NCBI RefSeq *Caudovirales* database (downloaded on 1 April 2020) was calculated using the following formula:$$wGRR_{A,B}=\underset{i}{\sum }\frac{id\left(A_{i}B_{i}\right)}{\min\left(A,B\right)}\times 100$$
where *id*(*A_i_B_i_*) is the sequence identity between each pair *i* of homologous proteins present in *A* and *B*, and min(*A*,*B*) is the number of proteins in *A* or *B* (whichever is smallest). This yields a score which represents the fraction of homologs in the shorter of the two bacteriophages weighted by sequence similarity. Homologous proteins were identified using diamond alignment searches, keeping bidirectional best hits with an E-value < 10^−5^. Each best hit for the predicted prophages (i.e., the closest reference genome based on *wGRR* match) was further investigated to confirm that additional appropriate characterisation had been carried out for the reference isolate to correctly assign its family. To calculate a *wGRR* cutoff for family assignation, the pairwise *wGRR* values between all bacteriophages in the NCBI *Caudovirales* database were calculated and the *wGRR* value (4.7) whereby two bacteriophages had ≥95% probability of belonging to the same family was determined.

### 2.4. In Silico Identification of Conserved, Previously Characterised Functional Lactobacillaceae Bacteriophage Sequences

Genome sequences of previously characterised functional bacteriophages that have been shown to have a permissive host (Appendix A) were obtained from the NCBI RefSeq database. Translated protein coding sequences from each bacteriophage genome were individually aligned against all protein-coding sequences in the *Lactobacillaceae* strain genomes using the blastp option in the NCBI standalone BLAST suite [60]. Protein hits with ≥80% query sequence coverage and an E-value ≤ 10^−5^ were retained for further analysis. Cases where ≥80% of the coding sequences present in a published phage genome were found in an individual *Lactobacillaceae* spp. genome were interrogated further to determine the location and synteny of matches. These regions were extracted and compared using Easyfig (version 2.2.2) [70].

### 2.5. In Silico Identification of Virulence Factors in Predicted Phage Sequences

To predict clusters of orthologous groups (COG) categories for putative prophage-encoded genes, all predicted coding sequences from the identified prophages were first clustered using cd-hit [71] with a sequence identity threshold of 0.9. Representatives from each cluster were compared to the eggNOG 4.5 database, and the predicted categories were extracted for each of the clusters. All (i.e., unclustered) predicted prophage coding sequences were also evaluated for virulence factors and potential antibiotic resistance determinants by performing BLAST searches against five databases: (i) Card [72] (protein homologue database), (ii) ResFinder [73] (the main ResFinder databases), (iii) AMRFinder [74] (amr and virulence factor database), (iv) VFDB [75] (core dataset, which only includes experimentally validated virulence factors); and (v) Victors [76] (virulence factor proteins). For all the databases, a threshold of >70% gene coverage and >80% identity was used to call a hit, to reduce the risk of identifying pseudogenes [77]. Finally, all predicted bacteriophage coding sequences were compared to the ResFams [78] hidden Markov models (HMMs) database (full database) using HMMER, version 3.3 (hmmer.org, accessed 18 November 2019) and an E-value cutoff of 10^−5^.

### 2.6. In Vitro Identification of Prophages in Vaginally-Derived and Commercially Available Probiotic Lactobacillaceae Genomes

Bacteriophages were induced from 39 *Lactobacillaceae* strains isolated from vaginal secretions of South African women and 18 *Lactobacillaceae* strains isolated from commercially available probiotics (see Appendix A for product details), as described previously [12]. Briefly, bacteria were grown anaerobically in MRS broth (Sigma-Aldrich^®^, St Louis, MO, USA) supplemented with 0.05 g/L L-Cysteine hydrochloride monohydrate (Sigma-Aldrich^®^, St Louis, MO, USA) and 0.1% Tween80 (Sigma-Aldrich^®^, St Louis, MO, USA) (MRSc). To induce bacteriophages, 0.4 µg/mL of mitomycin C (Sigma-Aldrich^®^, St Louis, MO, USA) was added to MRSc liquid cultures, as described previously [25]. Growth (OD_600nm_) was monitored over 18 h in cultures with and without mitomycin C (VersaMAX microplate reader, Molecular Devices, San Jose, CA, USA). After 18 h chloroform (10% *v*/*v*) was added to the mitomycin C-treated bacterial cultures, the cultures were centrifuged to remove bacterial debris (4500 g; Eppendorf centrifuge 5810, Eppendorf AG, Germany), and the supernatant filtered (0.2 µm, Minisart^®^ NML Syringe Filter, Type 17823, Sartorius Stedim Plastics GmbH, Göttingen, Niedersachsen, Germany). Bacteriophages were pelleted by centrifugation at 20,800 *g* (Eppendorf centrifuge 5430R, Eppendorf AG, Hamburg, Hamburg, Germany) for 1 h at 4 °C and resuspended in 100 µL of TEM buffer (0.1 M NH_4_-acetate, pH 7). This process was repeated twice, after which the pellet was resuspended in 50 µL of TEM buffer and stored at 4 °C. TEM grids (Square 200 mesh copper grids, AGG2200C, Agar Scientific, Stansted, Essex, UK) were prepared with a layer of carbon and then freshly deionised. The bacteriophage sample (10 µL) was incubated on the TEM grid for 10 min at room temperature. The grid was washed twice with 10 µL of H_2_O, and stained with 10 µL of 2% uranyl acetate (AGR1260A, Agar Scientific, Stansted, Essex, UK) for 3 min. A FEI Tecnai T20 (FEI Company, Hillsboro, Oregon, USA) was used to capture images.

### 2.7. Data Analysis

All data analyses, statistical calculations and plots were done in R (www.r-project.org; accessed on 1 July 2019). Shapiro–Wilk tests for normality were performed to determine the distribution of variables within the dataset. Planned descriptions of continuous variables with means, medians, standard deviations and proportions, as appropriate, were calculated. Overall differences in medians (nonparametric data) between species were tested for using the Kruskal–Wallis test, and the Dunn’s test was applied to adjust *p*-values for multiple comparisons. The Spearman’s rank test was applied to test for correlations between paired variables. Statistical *p*-values < 0.05 were considered statistically significant. Only *p*-values adjusted for multiple comparisons are reported.

## 3. Results

### 3.1. Detection of Putative Prophages in Publicly Available Lactobacillaceae Genomes

We analysed 897 high-quality genomes of the *Lactobacillaceae* family (Figure 1A, Appendix A) to determine the pervasiveness of prophages. Represented among these were 12 species of the genera *Lactobacillus*, *Lacticaseibacillus*, *Ligilactobacillus, Lactiplantibacillus* and *Limosilactobacillus.* We selected the following species as they are commonly found in the FGT and/or present in commercial probiotics marketed for FGT use: 30 *Lactobacillus acidophilus*, 70 *Lactobacillus crispatus*, 25 *Lactobacillus gasseri*, 49 *Lactobacillus helveticus*, 19 *Lactobacillus iners*, 16 *Lactobacillus jensenii*, 10 *Limosilactobacillus mucosae*, 344 *Lactiplantibacillus plantarum*, 116 *Limosilactobacillus reuteri*, 139 *Lacticaseibacillus rhamnosus*, 77 *Ligilactobacillus salivarius* and 2 *Limosilactobacillus vaginalis* genomes. In each species, we analysed all publicly available genomes. We opted to exclude *L. vaginalis* from further analyses due to the small number of available genomes, resulting in a final dataset of 895 genomes (Figure 1A).

We examined all genomes for evidence of integrated prophage sequences using five different prophage detection algorithms. The Prophet and Virsorter algorithms tended to identify higher numbers of prophages per genome across all *Lactobacillaceae* spp. compared to the Phaster, Phigaro and PhiSpy algorithms, with Phaster identifying the fewest (Appendix A). Phaster, PhiSpy and Virsorter tended to predict larger prophage genome sizes than Phigaro and Prophet (Appendix A), suggesting that in a substantial number of instances the boundaries of detected prophage sequences may have been differently identified by different algorithms.

All five algorithms identified prophages that are below the size range of bacteriophages in the NCBI *Caudovirales* database (<11.6 Kbp), which were excluded from further analysis since they were considered unlikely to be functional. PhiSpy predicted three prophages that were significantly larger than the largest known prophage genomes (>800 Kbp) and these were also excluded. Since variations in predicted prophage numbers, sizes and classifications were observed depending on the prophage detection algorithm used, we proceeded using two consensus datasets. A ‘lenient’ dataset consisted of all prophages identified by ≥2 algorithms with >80% reciprocal sequence overlap (*n* = 1671) and provided a plausible upper bound for our estimates. A ‘stringent’ dataset consisted of only those prophages that were detected by ≥4 of the algorithms with >80% reciprocal sequence overlap (*n* = 448) and provided a lower bound for our estimates. While 40–50% fewer prophages were detected with support from ≥4 algorithms than were detected with support from ≥2 algorithms, only 10% of the prophages detectable with ≥4 algorithms were detected by all five of the algorithms used (Figure 1B).

The 1671 prophages in the lenient dataset were predicted within 83.7% (742/895) of the examined *Lactobacillaceae* genomes, of which each contained between one and six prophage-like elements (Appendix A). In contrast, the 448 prophages in the stringent dataset were predicted within 40.1% (359/895) of the *Lactobacillaceae* strains, with a maximum of three prophage-like elements per strain being present (Appendix A).

### 3.2. In Vitro Confirmation of In Silico Predicted Prophage Prevalences

Chemical treatment of lysogenic strains with mitomycin C is known to cause induction of temperate bacteriophages [25]. We used this classic laboratory approach to induce prophages from 39 vaginally-derived *Lactobacillaceae* strains from 24 South African women and 18 *Lactobacillaceae* strains purified from eight commercial probiotic products marketed for FGT health in South Africa, France, or Spain (Table 1 and Appendix A) to confirm whether the in silico predicted prophage prevalence of 83.7% using the lenient and 40.1% using the stringent approach lies within the range that can be expected to be found in reality.

Twenty-eight of the 39 vaginally-derived isolates (71.8%) and 11/18 of the probiotic isolates (61.1%) displayed reduced growth in the presence of mitomycin C, suggesting possible induction of temperate bacteriophages. Eleven isolates (7/11 probiotic and 4/28 vaginally-derived *Lactobacillus* strains) showed >50% growth reduction in the presence of mitomycin C (Appendix A), and all these isolates contained bacteriophage particles in the supernatant, as determined by electron microscopy (Figure 2). However, some strains that did not show such a strong reduction in growth also had bacteriophage particles present in the supernatant, with 34/39 samples overall having bacteriophage particles present as confirmed by electron microscopy (Figure 2, Appendix A). These in vitro results suggest that 56.6% of the bacterial isolates carry inducible prophages, indicating that the ‘true’ prevalence of prophages in *Lactobacillaceae* members might be indeed within the range of our stringent and lenient in silico predictions.

### 3.3. Characterisation of Putative Prophages Detected in Publicly Available Lactobacillaceae Genomes

Within the stringent in silico dataset, *L. rhamnosus* genomes contained significantly more putative prophages than all other species besides *L. mucosae* and *L. jensenii* (Figure 3A, Appendix A). In addition, the proportion of strains harbouring prophages was higher for *L. mucosae* (7/10; 70%) and *L. rhamnosus* (96/139; 69.1%) than for the other species (0–49.1%). Bacterial genome size did not correlate with the number of predicted prophages (Spearman rho = 0.06, *p* = 0.8022). Similar observations were made within the lenient dataset, where in addition to *L. rhamnosus*, *L. reuteri* and *L. plantarum* genomes had a higher number of predicted prophages compared to most other *Lactobacillaceae* species (Figure 3A, Appendix A).

We next examined the size ranges of the predicted prophages (Figure 3B). Given the variability between the detection algorithms with respect to defining the prophage boundaries, we opted to be conservative and define the lengths of the prophages based on the algorithm that yielded the shortest length estimate. The sizes of the predicted prophages in the stringent dataset ranged from 14.09 to 97.19 Kbp, with putative prophages from *L. crispatus* genomes tending to be smaller than those from *L. gasseri* (*p* = 0.0455) and *L. plantarum* (*p* = 0.0373). Prophages from *L. plantarum* were significantly larger than those from *L. reuteri* (*p* = 0.0001) and *L. rhamnosus* (*p* = 0.0004) (Appendix A).

A weighted gene repertoire relatedness (*wGRR*) similarity scoring system [68,69] was used to putatively assign detected prophages to bacteriophage families when compared to the NCBI *Caudovirales* database (Figure 4A and Appendix A). While every prophage in our stringent dataset had a best hit wGGR score > 4.7 (as described in the method section, Appendix A) and could therefore be provisionally assigned to a family (based on sequence) (Figure 4B), a small number (20/1671; 1.2%) of prophages in the lenient dataset had best hit wGGR scores of <4.7 (Figure 4A). Almost half (47.7%) of the prophages in the stringent dataset had a best hit score of <30, with an even higher proportion (78.9%) of prophages in the lenient dataset having a hit score of <30, suggesting that they were only distantly related to bacteriophages represented in the NCBI database and might thus belong to uncharacterised bacteriophage species or genera. In both datasets most putative prophages were predicted to likely belong to the *Siphoviridae* family, with most of the remainder showing best hits to members of the family *Myoviridae* (Figure 4B and Appendix A).

### 3.4. Correlation of Number of Predicted Putative Prophages with Genome Quality

Given that sequencing coverage and assembly quality were likely to vary between the *Lactobacillaceae* spp. genomes that are present in the NCBI RefSeq database, and that prophages might be easier to detect in genomes that are of higher quality and more complete, we evaluated the relationship between both the genome ’completeness‘ (determined by checkM assessing the quality of a genome using a broad set of marker genes) and genome ’fragmentation‘ (estimated by scaffold N50, the shortest contig length needed to cover 50% of the genome) and the numbers of predicted prophages within those genomes. Across all the analysed genomes (Appendix A), but not on an individual species level (Appendix A), the numbers of predicted prophages per genome were significantly positively correlated with genome completeness for both the lenient (Spearman rho = 0.2; *p* < 0.0001) and stringent datasets (Spearman rho = 0.3; *p* < 0.0001). There was, however, no detectable correlation between numbers of predicted prophages and genome fragmentation in either the lenient or stringent datasets (Appendix A). This latter result is surprising because degrees of fragmentation differed significantly between the genomes of different *Lactobacillaceae* spp. (Appendix A), and completeness and fragmentation correlated positively (e.g., Spearman rho = 0.3958; *p* < 0.0001 in the stringent dataset). This suggests that the degree of completeness but not the degree of fragmentation of an individual genome influences the numbers of putative prophages that will be detectable within that genome, which is in agreement with previous observations that intact prophages tend to have few of the repeats within their genomes [79,80] that would result in genome fragmentation during the assembly of sequencing data.

### 3.5. Genome Completeness and Contamination Analysis of Identified Prophages

To gauge the completeness of predicted prophages, CheckV analyses were performed. Almost half of the predicted prophage genomes in both datasets (stringent-192/448, 42.86%; lenient-701/1671, 41.95%) were estimated to be ‘complete’ or ‘high-quality’ (genome likely to be >90% complete), indicating their likely functionality (Figure 5A). Of the complete and high-quality genomes in the stringent and lenient datasets, 38/192 (19.8%) and 304/701 (43.37%), respectively, were predicted to contain <10% host sequence contamination (Figure 5B,C), suggesting that most predicted prophages with high predicted degrees of genome completeness also had low degrees of host sequence contamination.

### 3.6. Identification of Conserved, Previously Characterised Functional Bacteriophage Sequences in Lactobacillaceae Genomes

To further investigate the functionality of the putative prophages, we compared the inferred amino acid sequences of their potential proteins to those of 51 previously characterized functional bacteriophages infecting members of the *Lactobacillaceae* family (Appendix A). Prophages with >80% amino acid sequence identity to known bacteriophages, and with >80% of the genome coverage of known bacteriophages, were considered as positive matches to the known bacteriophages. Three putative *L. jensenii* prophages matched bacteriophage Lv-1 (Figure 6A), one *L. plantarum* prophage matched the Sha1 bacteriophage (Figure 6B) and three *L. gasseri* prophages matched bacteriophage phi jlb1 (Figure 6C). No close matches were found between the remaining putative prophages and any of the 51 known functional bacteriophages infecting members of the *Lactobacillaceae* family. There were also no close matches to any of the known *Caudovirales*-associated bacteriophages represented within the RefSeq collection, indicating that most predicted prophages in our dataset are novel.

For all seven of the close matches, the functional organisation of genes was conserved between the published bacteriophage genomes and those genes inferred to be present in the putative prophages, although for both the Lv-1-like and Sha1-like prophages, some genome rearrangement was evident (Figure 6A,B). In the case of the Lv-1-like prophage, the rearrangement involved an inversion of the packaging, structural and lysis modules and the genetic switch and replication modules. For the Sha1-like prophages, the rearrangement involved the inversion of the replication/modification/regulation, packaging, structural, lysis and lysogeny modules. For the phi jlb1-like prophage, although the functional organisation was similar to that of phi jlb1, there was evidence of gene replacement within the packaging module in *L. gasseri* ATCC 33323 and *L. gasseri* TF08-1 relative to the phi jlb1 reference sequence (Figure 6C).

### 3.7. Identification of Virulence Factors and Antimicrobial Resistance Genes in Predicted Prophage Sequences

Bacteriophages are known to facilitate the horizontal transfer of genes [32]. To estimate whether any virulence and/or antimicrobial resistance genes are present within the genomes of the putative prophages, we predicted the functional categories of coding sequences contained in our dataset of putative prophages using eggNOG [81].

Within the stringent dataset, 95,575 putative coding sequences were identified and grouped into 3903 clusters by cd-hit [71]. More than two-thirds (68%) of these coding sequences either did not match any sequences in the eggNOG database or matched to curated orthologous group (COGs) of unknown function (COG category S). A further 23.3% matched to the following COG categories: replication, recombination and repair (11.2%; category L); transcription (6.4%; category K) and cell wall/membrane/envelope biogenesis (5.7%; category M). The remaining 13 COG categories were each represented by <1.5% of the predicted putative prophage genes.

Fifty-six putative prophage coding sequences were assigned to the defence COG category (category V), of which 16.1% (9/56) were assigned as homologues of Abi-like proteins (components of abortive infection responses) by the eggNOG database, 19.6% (11/56) were homologues of methyltransferase components of Type I restriction modification systems, and 3.6% (2/56) were ABC transporter homologues (previously shown to be upregulated in response to bile stress), one was a β-lactamase homologue (specifically of a serine hydrolase present in prophages from two *L. jensenii* strains), and the remaining 33 were homologues of various nucleases (a widespread component of the DNA packaging machinery).

Similar distributions were seen in the lenient dataset, where approximately 63% of the putative prophage protein coding sequences either did not match any sequences in the eggNOG database or matched COG category S. A further 23.92% of the predicted coding sequences matched COG categories L, K and M. Three COG categories were represented by >1.5% of the remaining predicted prophage genes: cell cycle control, cell division, chromosome partitioning (category D, 1.6%), amino acid transport and metabolism (category E, 1.9%) and defence (category V, 1.6%).

Similar to the stringent dataset, COG category V coding sequences were annotated as various nucleases (62/137, 45.3%), methyltransferases of Type I and Type II restriction modification systems (34/137, 24.5%), Abi-like proteins (20/137, 14.6%), ABC transporter homologues and permeases (11/137, 8.0%), ATPases (3/137, 2.2%) and a putative glycopeptide resistance gene, *vanZ*, in two predicted *L. rhamnosus* prophages. For the lenient dataset, putative β-lactamase genes were found in 33 predicted *L. plantarum* prophages, two *L. jensenii* prophages and a single predicted *L. gasseri* prophage.

To specifically assess whether the predicted prophages harboured any known virulence factors (including toxins) or genes with confirmed clinical antimicrobial resistance, we examined them using the ResFams, Card, Resfinder, AMRFinder, VFDB and Victors databases. We identified putative ABC efflux pumps (all most similar to ResFam RF0007) in 10/448 putative prophage genomes (2.5%, all located in *L. plantarum* genomes) in the stringent dataset and in 125/1691 (7.4%) putative prophages of the lenient dataset. One additional hit for a potential aminoglycoside transferase (ResFam RF0033, possible providing aminoglycoside resistance) was predicted in one *L. gasseri* prophage in the lenient dataset.

Given the potential roles for these genes in antimicrobial resistance, the predicted β-lactamase, the putative glycopeptide resistance gene *vanZ*, the ABC efflux pump and the aminoglycoside transferase genes were investigated for their possible functionality. While the putative ABC efflux pump contained domains typical of ABC-type transporters (transmembrane domains, Walker A and B motifs, ABC signature sequence), similarity searches against the curated Swiss-Prot database revealed low similarity (42.6% amino acid identity) to previously characterised fatty acid transporter proteins from *Mycobacterium tuberculosis*. Additionally, alignment with the well-characterised lactococcal ABC multidrug efflux pump LmrC yielded an even lower degree of similarity (30.9% amino acid identity) (Appendix A).

Similarly, the translated β-lactamase contained peptidase domains previously identified in class C β-lactamases, but alignment of the predicted amino acid sequence with previously characterised functional AmpC proteins revealed only a low degree of similarity (<30% amino acid sequence identity, Appendix A).

The translated aminoglycoside transferase contained the APH domain common to proteins that confer aminoglycoside resistance. This gene has been identified in a previous screen for antibiotic resistance genes in the family *Lactobacillaceae* but has not yet been functionally characterised [58].

Finally, the translated VanZ product shared 32.4% amino acid identity with the VanZ in *Enterococcus faecium* BM4147 (Appendix A). This protein has been shown to confer low-level resistance to teicoplanin but not vancomycin in *E. faecium* [82]. The *vanZ* gene was present in all *L. rhamonusus* genomes included in this study. However, unlike the *vanZ* in *E. faecium*, which is located on Tn*1546*, in *L. rhamnosus* the gene did not appear to be part of a transposon.

Closer inspection of the genomic context for the open reading frames (ORFs) encoding all four putative AMR genes indicated that they were located near to predicted host/prophage genome boundaries, raising the possibility that they were features of host genomes that had been misclassified as belonging to prophage sequences. Therefore, we examined the four genetic loci in *Lactobacillaceae* strains with and without a predicted prophage at each locus. For all four putative antimicrobial resistance factors, identical proteins were present in highly homologous regions in strains with and without putative prophages, suggesting that they were more likely to be components of the host genome (Appendix A). For the locus containing the putative efflux pump, the prophages (all predicted to belong to the *Siphoviridae* family) shared a relatively low degree of overall similarity, suggesting that different bacteriophages may have independently integrated at similar sites in relation to what may in fact be to be a native *Lactobacillaceae* ABC efflux pump gene. Similar results were seen for the locus containing the annotated *vanZ* gene, although the two prophages (both potential siphoviruses) did share some sequence similarity in what appeared to be their replication modules (based on the presence of recombinase and DNA replication genes). In contrast, the prophages integrated near the putative β-lactamase and aminoglycoside transferase encoding loci in different *Lactobacillaceae* genomes were identical and located immediately downstream of tRNA genes (tRNA-Glu/tRNA-Gln and tRNA-Leu, respectively), a known hotspot for prophage integration.

## 4. Discussion

Probiotics containing *Lactobacillus* spp. and other species of *Lactobacillaceae* genera are gaining increasing public backing as therapeutics for restoring and/or increasing the stability of human gut and FGT microbiomes [16]. However, some commercial live therapeutics have been shown to harbour ARGs [41,42], and horizontal transfer of ARGs has been reported experimentally [33,34,35], in vivo [36,37,38] and via food consumption [40]. Probiotic candidates can also harbour prophages [19,24,25,26], and bacteriophages have been shown to contribute to the horizontal transfer of ARGs, also in animal models [33,44,45]. Although this has not been confirmed yet in humans [77], these previous studies raise the question whether bacteriophage-mediated HGT may lead to the transfer of ARGs from probiotic strains to resident bacteria [44,45], which could potentially pose a risk both to the FGT microbiota and to the human hosts of these microbiota. In an effort to generate plausible upper and lower bounds for our estimates of prophage prevalence among *Lactobacillaceae* spp., we used in silico prediction of prophage sequences at varying degrees of stringency to infer that between 40% and 80% of *Lactobacillaceae* genomes harbour at least one prophage. Accordingly, we were able to induce prophages from 60% of the *Lactobacillaceae* strains that we examined in vitro, suggesting that the true prevalence of prophages in members of these genera might lie somewhere within the limits of our stringent and lenient datasets.

### 4.1. High Uncharacterised Lactobacillaceae Prophage Diversity

The vast majority of the putative prophages that we identified within *Lactobacillaceae* genomes are uncharacterised and would likely be assigned to new species or genera if classified under the existing taxonomy. Even within the ‘stringent’ dataset, only half of the putative prophages in this high-confidence dataset had best hit wGGR scores of <30, implying that the bacteriophages from which these prophages were derived were only distantly related to the known bacteriophage species that are represented within the NCBI RefSeq database. Many of the prophages not identified by at least four of the detection algorithms are likely genuine prophages that were not more confidently identified as such simply because they are too different to currently known bacteriophages. Currently described *Lactobacillaceae* bacteriophages primarily belong to the *Siphoviridae* family, although representatives from the *Myoviridae* and *Herelleviridae* have also been described [19,83]. Our in silico predictions based on best hit wGGR score supported this. However, it is important to note that prophage identification by sequence analysis alone is not sufficient to confidently assign family-level classifications and additional experimental evidence, including induction and bacteriophage particle characterisation is required to validate these results.

### 4.2. Ubiquitous Presence of Prophages among Lactobacillaceae spp.

As prophages were detectable in at least one strain of each species, it would be problematic to deem any genera or species of *Lactobacillaceae* as being prophage-free based solely on their taxonomic assignments: a factor that will complicate the development of guidelines for prioritizing the use of any specific *Lactobacillaceae* species for bacteriophage-free probiotics. A recent in silico screen of *Lactobacillus* genomes reported that *L. iners* harboured significantly fewer prophages than other vaginal *Lactobacillus* spp. [84], and the authors suggested a possible protective role for the *hsdR* gene which encodes a type I restriction enzyme and is commonly and uniquely present in *L. iners* genomes. A similar trend was seen in our analyses, as only 15% of the *L. iners* genomes harboured prophages when using our stringent identification approach. Even when using our lenient identification approach, *L. iners* genomes harboured prophages less commonly than most other species. The clinical relevance of these findings is questionable as, to our knowledge, no efforts are currently underway to include *L. iners* strains as probiotics for vaginal health.

### 4.3. Prophage Safety

We only found four potentially prophage-encoded genes (β-lactamases, ABC efflux pumps, glycopeptide resistance genes and aminoglycoside transferases) that might impact the safety of using the sequenced *Lactobacillaceae* strains as probiotics. However, it is unclear whether the detected prophages could indeed mediate the horizontal transfer of these genes. In fact, in all cases where these genes were detected in association with prophage sequences, it seems most likely that the prophages inserted near these conserved host genes, as identical/nearly identical genes were commonly present within bacterial strains without prophages. These results emphasise that, even with the use of five different prophage detection algorithms, we are still limited with respect to accurately defining the bounds of prophage sequences against a background of host genome sequences, as already acknowledged for one of the tools that we used [85]. Interestingly, some of the putative prophages detected in our study seemed to be inserted in close proximity to tRNA genes, which are well conserved in all organisms [86] and have previously been identified as preferred target sites for prophage insertion [87].

Interestingly, the ABC efflux pump gene was present in almost all *L. rhamnosus* strains evaluated here but was absent from most other *Lactobacillaceae* strains, indicating that this gene might be part of the *L. rhamnosus* core genome: a factor that might explain the ubiquitous presence of *L. rhamnosus* at various body sites [88], as bacterial ABC efflux pumps are known to be involved in the efflux of a variety of substrates other than antibiotics [89,90]. When evaluating the antibiotic susceptibility of vaginal *L. jensenii* strains that also harbour a β-lactamase, we found that all were sensitive to penicillin [12], indicating that the β-lactamase might not be a functional ARG. Further experimental studies are therefore needed to better characterise the identified putative resistance genes to determine whether they are indeed functional and transferrable. It also needs to be acknowledged that if it can be ensured that no virulence factors are present within a given prophage sequence, then the presence of that prophage sequence within a probiotic *Lactobacillaceae* strain might be advantageous due to the superinfection immunity and subsequent growth advantages that the prophage could confer on that strain [47].

Limitations of our study include the biased composition of the *Lactobacillaceae* whole genome dataset that we analysed, as *L. rhamnosus, L. reuteri* and *L. plantarum* were significantly over-represented in our datasets, and we did not manually curate prophage-host boundaries. In addition, our in silico analysis relied on publicly available genome sequences housed in the NCBI Refseq database and, while these sequences are subject to manual curation and we included some additional quality control checks, they may still have contained sequence errors. Finally, we acknowledge that using mitomycin C to induce prophages in vitro does not accurately reflect conditions in the in vivo environment of the FGT. Nevertheless, given that we were able to identify putative prophages in up to 80% of *Lactobacillaceae* strains in silico and up to 60% in vitro, it is comforting to know that few of the putative prophages potentially encode known virulence factors, toxins or antibiotic resistance genes. Further biological characterisation of the induced prophages, including host range testing and genome sequencing, would allow for more detailed description and classification of the induced bacteriophages.

## 5. Conclusions

In summary, our results suggest that *Lactobacillaceae* prophages are unlikely to pose any risk to humans, yet support recent recommendations by the European Food Safety Authority to perform whole genome sequencing analyses on microorganisms that are intentionally used in the food chain as part of a comprehensive assessment of their safety [91]. We further propose that the presence of potential prophages (and/or virulence genes, effectors and/or ARGs) should be taken into consideration when selecting probiotic candidates for commercialisation [58]. Translating in silico predictions to clinically relevant functional phenotypes remains a challenge, and the functionality and host ranges of putative prophages may still need to be verified experimentally both in vitro and ex vivo.

## Figures and Tables

**Figure 1 microorganisms-10-00214-f001:**
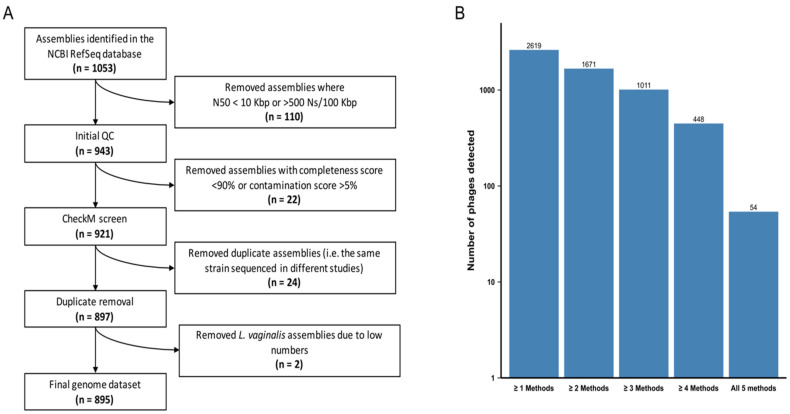
Generating *Lactobacillaceae* genome and prophage datasets. (**A**) Path to achieve a final *Lactobacillaceae* genome sequence dataset. The original publicly available dataset was filtered by taking into account genome fragmentation (estimated by scaffold N50 and number of ambiguous bases) and genome completion (estimated by CheckM). (**B**) Number of putative bacteriophages predicted by five different algorithms: Prophet, Virsorter, Phaster, PhiSpy and Phigaro. Bacteriophages predicted by ≥2 and ≥4 algorithms were retained in the ‘lenient’ and ‘stringent’ datasets, respectively.

**Figure 2 microorganisms-10-00214-f002:**
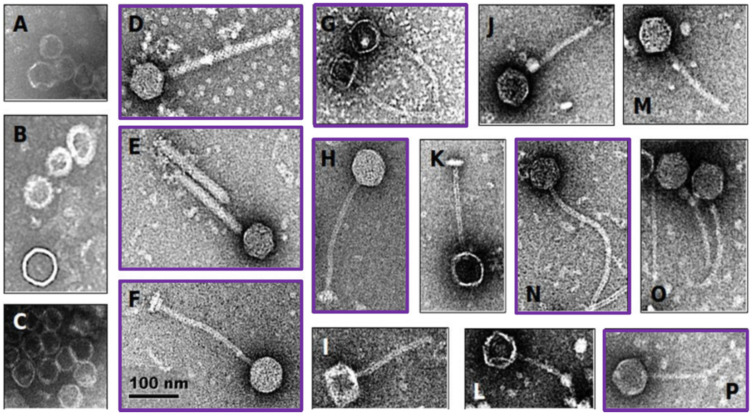
Electron microscopy of induced *Lactobacillaceae* bacteriophages. Probiotic and vaginally-derived *Lactobacillaceae* isolates were treated with mitomycin C to induce bacteriophages. The majority contained capsid-like structures (examples shown in **A**–**C**) and others contained head-tail structures (examples shown in **D**–**P**). The scale applies to all panels of the figure. Purple boxes indicate bacteriophages induced from probiotic *Lactobacillaceae* isolates, while the remaining ones were induced from vaginally-derived *Lactobacillaceae* isolates.

**Figure 3 microorganisms-10-00214-f003:**
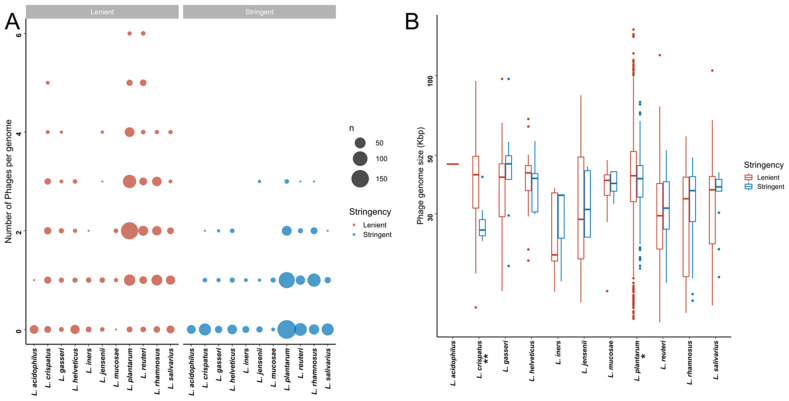
Putative bacteriophages that are detectable within publicly available *Lactobacillaceae* genome sequences. (**A**) Number of putative bacteriophages per *Lactobacillaceae* genome in the lenient (red) and stringent (blue) datasets. The size of each dot is proportional to the total number of strains with the indicated number of putative bacteriophages. (**B**) Genome sizes of the detected bacteriophages genome in the lenient (red) and stringent (blue) datasets. * indicates that the adjusted *p*-value was <0.05, and ** indicates that the adjusted *p*-value was <0.01.

**Figure 4 microorganisms-10-00214-f004:**
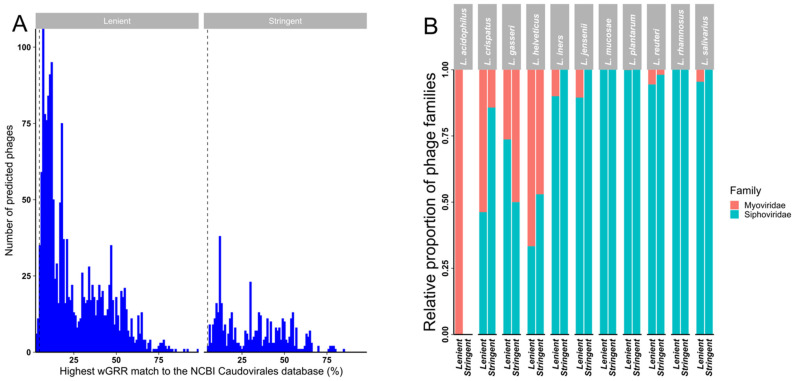
Weighted gene repertoire relatedness (wGGR) for predicted prophages in the lenient and stringent datasets. (**A**) The highest hits *wGRR* match of the predicted bacteriophages to bacteriophages in the NCBI *Caudovirales* database in the lenient and stringent datasets. (**B**) Relative proportions of likely bacteriophage families in the lenient and stringent datasets calculated using the *wGRR* approach by *Lactobacillaceae* spp.

**Figure 5 microorganisms-10-00214-f005:**
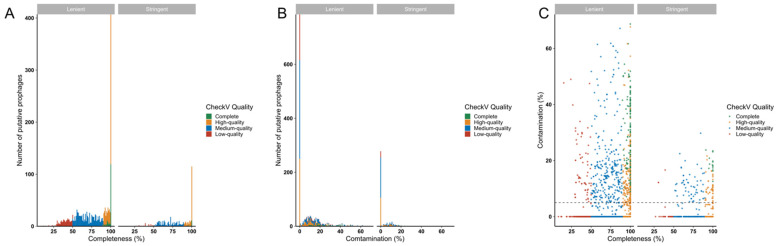
Estimating likely functionality of prophages detected in *Lactobacillaceae* genomes using CheckV. The quality, including completeness (**A**,**C**) and contamination (**B**,**C**) of the predicted prophages, was evaluated using CheckV. Prophages were colour-coded according to their quality, including complete (green), high-quality (>90% complete, yellow), medium-quality (50–90% complete, blue) and low quality (<50% complete, red) prophages.

**Figure 6 microorganisms-10-00214-f006:**
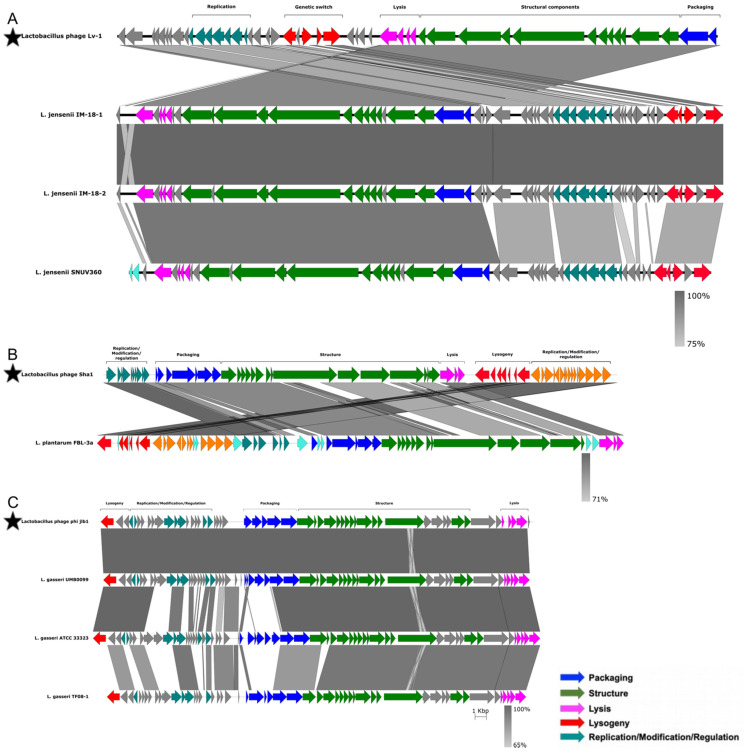
Analysis of similarity and organisation of prophages detected in *Lactobacillaceae* genomes compared to previously characterised functional *Lactobacillaceae* bacteriophages. The sequences of previously characterised *Lactobacillaceae* bacteriophages are indicated with stars. Lv-1 (**A**), Sha1 (**B**) and phi jlb1 (**C**) were aligned to the putative prophages from our dataset. Genes were annotated and colour-coded based on their functions, with the degree of similarity between previously described bacteriophages and putative prophages being indicated by different shades of grey. Original terminology was used for each bacteriophage from the respective manuscripts.

**Table 1 microorganisms-10-00214-t001:** List of probiotic and female genital tract (FGT)-derived isolates included in the study.

Species	*n*	Isolated From:
*L. crispatus*	10	FGT (113.22PA, 100.16PA, 73.55a, 96.9PB, 100.16a, 94.97PA, 70.6PA, 80.3a, 94.77 PA, 95.34 PA)
	1	Jarro Dophilus^®^ Women (Jarrow Formulas, Inc., Los Angeles, CA, USA)
	1	Physioflor Vaginal Flora Natural Probiotic (Laboratoires Iprad, Paris, France)
	1	Vacramal^®^ (Nutriphyt NV, Beernem, Belgium)
	1	Probiovance^®^ intim (Ysonut Laboratories, Barcelona, Spain)
*L. gasseri*	7	FGT (117.73PA, 94.98PB, 100.46PA, 107.10PA, 114.1PA, 114.12PA, 119.1PA)
*L. jensenii*	11	FGT (113.22PA, 92.1PA, 88.10PA, 88.33PA, 94.70 PA, 95.1 PA, 95.22 PA, 84.35 PA, 73.2 PA, 96.8 PA, 96.45 PA)
*L. vaginalis*	5	FGT (80.23b, 88.5b, 91.8a, 100.13 PA, 79.24 PA)
*L. mucosae*	6	FGT (90.13PA, 80.23a, 99.1 PA, 85.1 PA, 87.5 PA, 98.46 PA)
*L. rhamnosus*	1	Lactogyn^®^ (Jadran-galenski laboratorij, Rijeka, Croatia)
	2	Vagiforte^®^ (Bioflora CC, Centurion, South Africa)
*L. reuteri*	1	Probiovance^®^ intim (Ysonut Laboratories, Barcelona, Spain)
	2	Vagiforte^®^ (Bioflora CC, Centurion, South Africa)
*L. acidophilus*	1	MediGYNE^®^ (Laboratoires IPRAD, Paris, France)
	1	BactiGyn^®^ (Laboratoire CCD, Paris, France)
	2	Vagiforte^®^ (Bioflora CC, Centurion, South Africa)
*L. plantarum*	1	BactiGyn^®^ (Laboratoire CCD, Paris, France)
	1	Vacramal^®^ (Nutriphyt NV, Beernem, Belgium)
*L. salvarius*	1	BactiGyn^®^ (Laboratoire CCD, Paris, France)

## Data Availability

Publicly available datasets were analysed in this study. These data can be found at https://www.ncbi.nlm.nih.gov/assembly (Access date: 1 June 2019).

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
