# Peer review of "In Silico Characterisation of Putative Prophages in Lactobacillaceae Used in Probiotics for Vaginal Health"

_microorganisms, 2022, doi:10.3390/microorganisms10020214_

Round 1

Reviewer 1 Report

The manuscript describes presence of bacteriophages in publically available Lactobaciallaceae genomes. Four different algorithm for prophage identification in genomes were used and compared, this is very valuable approach as no precise method for all bacteria/prophages is currently available. Bioinformatics analyses are supplemented with in vitro testing of bacteriophage induction. The manuscript brings some interesting data, I have some suggestions for its improvements and after incorporating them I recommended it for publication:

- (page 5, line 208) I suppose that bacteriophages were precipitated by addition of some precipitating agent (PEG?) and not directly from the medium or TEM buffer

- (page 7, line 271) the results of prophage induction from probiotic strains and vaginal isolates should be presented in more details. Was correlation between mitomycin growth inhibition and TEM results observed? The results for particular strains should be added as supplementary table. The identity of induced phages should be analyzed (especially in strains with known genome sequences) or at least this possibility should be mentioned in discussion.

- I suppose that mutual relativeness of all detected prophages were calculated by wGRR approach and the results are presented in Fig. S2. This figure is quite interesting, it should be presented in main text and commented. Identified prophages should be clustered to groups bellow family and frequency of these groups shoud be analyzed in species – are relative prophages present in different species?

- (page 10, line 356) fig. 5 (showing comparison of prophages with reference phages) is missing – it should be added into manuscript

- (page 14, line 564) description of supplementary Tables and Figures is insufficient, more information about presented data should be included (in main text or in supplementary material)

- reference 62 should be cited: Starikova, E.V., Tikhonova, P.O., Prianichnikov, N.A., Rands, C.M., Zdobnov, E.M., Ilina, E.N., Govorun, V.M. Phigaro: High-throughput prophage sequence annotation (2020) Bioinformatics, 36 (12), pp. 3882-3884.

Author Response

The manuscript describes presence of bacteriophages in publically available Lactobaciallaceae genomes. Four different algorithm for prophage identification in genomes were used and compared, this is very valuable approach as no precise method for all bacteria/prophages is currently available. Bioinformatics analyses are supplemented with in vitro testing of bacteriophage induction. The manuscript brings some interesting data, I have some suggestions for its improvements and after incorporating them I recommended it for publication:

We thank the reviewer for the constructive comments and suggestions have attempted to incorporate these into the revised manuscript.

- (page 5, line 208) I suppose that bacteriophages were precipitated by addition of some precipitating agent (PEG?) and not directly from the medium or TEM buffer

The bacteriophages were precipitated by centrifugation as described in the manuscript. No PEG or other precipitating agent was added.

- (page 7, line 271) the results of prophage induction from probiotic strains and vaginal isolates should be presented in more details. Was correlation between mitomycin growth inhibition and TEM results observed? The results for particular strains should be added as supplementary table. The identity of induced phages should be analyzed (especially in strains with known genome sequences) or at least this possibility should be mentioned in discussion.

Thank you, we have added these results as supplementary table (Table S5) and amended the result (lines 305-311) and discussion sections accordingly (lines 650-652).

305-311            “Eleven isolates (7/11 probiotic and 4/28 vaginally derived Lactobacillus strains) showed >50% growth reduction in the presence of mitomycin C (Table S5), and all these isolates contained bacteriophage particles in the supernatant, as determined by electron microscopy (Figure 2). However, some strains that did not show such a strong reduction in growth also had bacteriophage particles present in the supernatant, with 34/39 samples overall having bacteriophage particles present as confirmed by electron microscopy (Figure 2, Table S5).”

650-652            “Further biological characterization of the induced prophages, including host range testing and genome sequencing, would allow for more detailed description and classification of the induced bacteriophages.”

- I suppose that mutual relativeness of all detected prophages were calculated by wGRR approach and the results are presented in Fig. S2. This figure is quite interesting, it should be presented in main text and commented. Identified prophages should be clustered to groups bellow family and frequency of these groups shoud be analyzed in species – are relative prophages present in different species?

The reviewer is correct that we calculated mutual relatedness of all prophages in both the lenient and stringent datasets.  In line with their comments, we have updated the manuscript by including a dedicated figure (Figure 4) to report the results of the wGRR analysis. Unfortunately, since taxonomic information below family level is lacking for many of the sequences present in the NCBI Caudovirales database, it was not possible to determine a wGRR cutoff value to confidently assign our predicted prophages further. Given the additional points raised by reviewer 2 concerning the limitations of using sequence homology to determine prophage family, we have chosen not to carry out a statistical comparison between the relative abundance of the different Caudovirales family members in the Lactobacillaceae included in this analysis.

- (page 10, line 356) fig. 5 (showing comparison of prophages with reference phages) is missing – it should be added into manuscript

We have added the figure to the revised manuscript (Figure 6 based on the revised numbering).

- (page 14, line 564) description of supplementary Tables and Figures is insufficient, more information about presented data should be included (in main text or in supplementary material)

This information has been added to the supplementary material and manuscript (lines 660-730).

- reference 62 should be cited: Starikova, E.V., Tikhonova, P.O., Prianichnikov, N.A., Rands, C.M., Zdobnov, E.M., Ilina, E.N., Govorun, V.M. Phigaro: High-throughput prophage sequence annotation (2020) Bioinformatics, 36 (12), pp. 3882-3884.

Thank you for the correction, this reference has been updated in the revised manuscript (lines 917-919).

Reviewer 2 Report

General comment

It is a well-done work that deserves to be published by this journal after the below major issues have been addressed.

Major comment

The authors need to state what informed the decision to consider some of the prophages isolated as members of Myoviridae family.

It is important that the authors consider doing an in vitro analysis of these prophages’ genetic materials (extract their DNA and sequence them). In addition, they may do proteomic analysis of these phages protein to get more data for determining these phages’ classification.

Parts that needs explanation\revision:

 2.0 Materials and Methods

2.5. In silico identification of virulence factors in predicted phage sequences

L194: What made you choose this E-value as the cut-off point?

2.6. In vitro identification of prophages in in vaginally derived and commercially available probiotic Lactobacillaceae genomes                   

L208: Please state the model of the centrifuge used, the company that made it and the country of origin.

L210: Please state the source of the grid, (manufacturing company, town & country)/ the product’s catalogue number.

2.7. Data analysis

L223: Italicize probability (P) – value.

 3.0 Results

3.1. Detection of putative prophages in publicly available Lactobacillaceae genomes

L253-255: These families (Myoviridae and Podoviridae) are consist of only obligate lytic phages. Please provide sufficient reasons that made you consider these prophages as members of either Myoviridae or Podoviridae family. Phaster and Phigaro tools aren’t sufficient to ascertain the classification of a phage.

3.2. In vitro confirmation of in silico predicted prophage prevalence

L287-289: This statement is mis-placed (Figure 2: … contain phage that might belong to the Family Myoviridae). Electron micrographs D & E don’t depict any phage with a proper contractile tail (the photos should illustrate phages at non-contractile and contractile state). Therefore, it is not right to hypothesis that these phages are myoviruses.

L288, 289 & 290: Lactobacillaceae, Myoviridae & Siphoviridae are scientific terms which needs to italicized whenever applied/used in any article.

3.3. Characterisation of putative prophages detected in publicly available Lactobacillaceae genomes

Figure 3. …(C) Relative proportions of bacteriophage families calculated using the wGRR approach... Needs revision since the grouping is not in line with the ICTV classification guideline (https://talk.ictvonline.org/). Therefore, graph is misleading.

L313-314:

      1. NCBI data only is never conclusive when it comes to classification of phages since a large chunk of its data are full of errors.
      2. In addition, the analysis was not done with these phages’ sequence data (you haven’t shown anywhere in this manuscript how you got the original phages’ sequence). Sequence data derived from the global GeneBank (like NCBI etc) are occasionally full of errors and can’t be a true representative of a given isolated phage genome.
      3. You were to compare the original sequence data of these prophages (sequence of isolated phages) with what is in the global Gene-banks.

L323: … with most of the remainder belonging to the family Myoviridae (Figure 3C, Figure S2C)

This statement is misleading. Please do away with it (due to previously given reasons [L313-314]). You need the original sequence, proteomic and finally the micrographs to ascertain the classification of a phage isolate.

3.6. Identification of conserved, previously characterised functional bacteriophage sequences in Lactobacillaceae genomes

 L 364, 374 & 381: Please attach Figures 5A-C, they are missing in the main manuscript.

3.7. Identification of virulence factors and antimicrobial resistance genes in predicted prophage sequences

The results in this section are invalid since you never isolated & sequenced these phages’ DNA but opted for in silico way (used sequence data in the global GeneBank), which are rarely reliable.

What makes you think that the genome sequences in the global GeneBank represents the isolated phages' true genomes? 

Author Response

General comment

It is a well-done work that deserves to be published by this journal after the below major issues have been addressed.

We thank the reviewer for the constructive comments and suggestions have attempted to incorporate these into the revised manuscript.  Please find a point-by-point response below.

Major comment

The authors need to state what informed the decision to consider some of the prophages isolated as members of Myoviridae family.

It is important that the authors consider doing an in vitro analysis of these prophages’ genetic materials (extract their DNA and sequence them). In addition, they may do proteomic analysis of these phages protein to get more data for determining these phages’ classification.

While we agree with the reviewer that further characterisation of the isolated bacteriophages is important, this was not the main focus of the current study. The induction experiments were carried out to evaluate if the predictions obtained in the in silico portion of the analysis were biologically relevant, namely that between 40-80% of members of the Lactobacillaceae harbour prophages. The fact that we were able to induce prophages from 56% of isolates supports these predictions. Since the main argument of our manuscript is not the description of the induced prophages but rather the prevalence of prophages in Lactobacillaceae and in line with both reviewers’ suggestions, we have removed references to the possible identities of the induced prophages and updated the discussion to acknowledge the need for further characterisation (lines 590-594, 650-652).

590-594            “Our in silico predictions based on best hit wGGR score supported this. However, it is important to note that prophage identification by sequence analysis alone is not sufficient to confidently assign family-level classifications and additional experimental evidence, including induction and bacteriophage particle characterisation is required to validate these results.”

650-652            “Further biological characterization of the induced prophages, including host range testing and genome sequencing, would allow for more detailed description and classification of the induced bacteriophages.”

Parts that needs explanation\revision:

 2.0 Materials and Methods

2.5. In silico identification of virulence factors in predicted phage sequences

L194: What made you choose this E-value as the cut-off point?

We chose this E-value in keeping with our conservative approach to reduce the incidence of false positives.  The 1E-05 cut-off is based on the more stringent approach adopted by Enault et al (2017).

2.6. In vitro identification of prophages in vaginally derived and commercially available probiotic Lactobacillaceae genomes                   

L208: Please state the model of the centrifuge used, the company that made it and the country of origin.

We have added this information to the revised manuscript (line 213).

L210: Please state the source of the grid, (manufacturing company, town & country)/ the product’s catalogue number.

We have added this information to the revised manuscript (lines 216-217).

2.7. Data analysis

L223: Italicize probability (P) – value.

We have corrected this in the revised manuscript (e.g. lines 228, 230).

3.0 Results

3.1. Detection of putative prophages in publicly available Lactobacillaceae genomes

L253-255: These families (Myoviridae and Podoviridae) are consist of only obligate lytic phages. Please provide sufficient reasons that made you consider these prophages as members of either Myoviridae or Podoviridae family. Phaster and Phigaro tools aren’t sufficient to ascertain the classification of a phage.

We thank the reviewer for the useful comments and agree that Phaster and Phigaro alone are not sufficient to assign the identified prophages to families. Given that the wGRR approach represents a more robust approach (the limitations noted below notwithstanding), we have removed the Phaster and Phigaro family predictions from the manuscript (line 268). To acknowledge the limitations of the wGRR approach we have used “likely” or “putative” when referring to the their wGRR classification as Myoviridae or Siphoviridae throughout the manuscript.

3.2. In vitro confirmation of in silico predicted prophage prevalence

L287-289: This statement is mis-placed (Figure 2: … contain phage that might belong to the Family Myoviridae). Electron micrographs D & E don’t depict any phage with a proper contractile tail (the photos should illustrate phages at non-contractile and contractile state). Therefore, it is not right to hypothesis that these phages are myoviruses.

We have removed references to possible isolated phage families from the revised manuscript (line 338).

L288, 289 & 290: Lactobacillaceae, Myoviridae & Siphoviridae are scientific terms which needs to italicized whenever applied/used in any article.

This has been corrected throughout the manuscript.

3.3. Characterisation of putative prophages detected in publicly available Lactobacillaceae genomes

Figure 3. …(C) Relative proportions of bacteriophage families calculated using the wGRR approach... Needs revision since the grouping is not in line with the ICTV classification guideline (https://talk.ictvonline.org/). Therefore, graph is misleading.

We would appreciate clarity on this comment. Does this refer to the new ICTV taxonomy release earlier this year as well as the one implement in GenBank? For the wGRR analysis, we used only the complete Caudovirales reference genomes present in the NCBI RefSeq database, and we investigated each best hit (the closest related reference genome based on wGRR match) for our predicted prophages to confirm that appropriate characterisation had been carried out for the reference isolate and its family had been correctly assigned in each case.

L313-314:

      1. NCBI data only is never conclusive when it comes to classification of phages since a large chunk of its data are full of errors.
      2. In addition, the analysis was not done with these phages’ sequence data (you haven’t shown anywhere in this manuscript how you got the original phages’ sequence). Sequence data derived from the global GeneBank (like NCBI etc) are occasionally full of errors and can’t be a true representative of a given isolated phage genome.
      3. You were to compare the original sequence data of these prophages (sequence of isolated phages) with what is in the global Gene-banks.
  1. For the wGRR analysis, we used only the complete Caudovirales reference genomes present in the NCBI RefSeq database. We acknowledge that although genome sequences in this database are subject to manual curation (unlike Genbank) and mostly reflect genomes from well-characterised bacteriophages, errors with regards to family assignation may occasionally be present. Therefore, we investigated each best hit (n = 81) for our predicted prophages (i.e. the closest related reference genome based on wGRR match) to confirm that additional appropriate characterisation had been carried out for the reference isolate to correctly assign its family in each case. We have updated the manuscript to include this (lines 163-165).

163-165 “Each best hit for the predicted prophages (i.e. the closest reference genome based on wGRR match) was further investigated to confirm that additional appropriate characterisation had been carried out for the reference isolate to correctly assign its family.”

  1. We did not mean to imply that we used the wGRR approach to propose any family assignations of our induced bacteriophages. Those were based on phage particle morphology and as mentioned above we have removed any family assignation of the induced bacteriophages from the revised manuscript.
  2. For the Lactobacillaceae genomes we also only used sequences from the NCBI Refseq database (not Genbank), which undergoes a level of manual curation. We performed some additional QC checks but agree with the reviewer that the assemblies may still contain sequence errors. We have acknowledged this limitation more explicitly in the revised manuscript (lines 641-644).

641-644 “In addition, our in silico analysis relied on publicly available genome sequences housed in the NCBI Refseq database and, while these sequences are subject to manual curation and we included some additional quality control checks, they may still have contained sequence errors.”

L323: … with most of the remainder belonging to the family Myoviridae (Figure 3C, Figure S2C)

This statement is misleading. Please do away with it (due to previously given reasons [L313-314]). You need the original sequence, proteomic and finally the micrographs to ascertain the classification of a phage isolate.

We have revised the manuscript by removing family assignation of the induced bacteriophages and acknowledged in the discussion (lines 590-594) the limitations mentioned above.

590-594            “Our in silico predictions based on best hit wGGR score supported this. However, it is important to note that prophage identification by sequence analysis alone is not sufficient to confidently assign family-level classifications and additional experimental evidence, including induction and bacteriophage particle characterisation is required to validate these results.”

3.6. Identification of conserved, previously characterised functional bacteriophage sequences in Lactobacillaceae genomes

 L 364, 374 & 381: Please attach Figures 5A-C, they are missing in the main manuscript.

We have included these figures (now Figure 6A-C) in the revised manuscript.

3.7. Identification of virulence factors and antimicrobial resistance genes in predicted prophage sequences

The results in this section are invalid since you never isolated & sequenced these phages’ DNA but opted for in silico way (used sequence data in the global GeneBank), which are rarely reliable.

What makes you think that the genome sequences in the global GeneBank represents the isolated phages' true genomes? 

As noted in our previous answers, we agree with the reviewer that assembly data in the Refseq databases may contain sequencing errors although genome sequences are subject to some manual curation (unlike Genbank). Moreover, as acknowledged in our discussion, even though candidate virulence factors and antimicrobial resistance genes were first identified by algorithms within the predicted prophage/host genome, we showed (Figure S4) that most of the genes were located near or on the boundary of prophages and, therefore, may not be part of the true phage genomes. We feel that the results of the virulence factor and AMR gene screens are still worth reporting given that so few candidate genes were identified and we have emphasised the need to carry out further in vitro experiments to validate these predictions in the revised manuscript (lines 632-634, 642-645) . Moreover, since the subject of AMR genes being horizontally transferred by bacteriophages is a current hot debate and mostly analysed through bio-informatic tools (see e.g. Modi et al 2013 Nature, Enault et al 2017 ISME J, Billaud et al 2021 ISME Com), we think that these sort of “negative results” are participating in this ongoing discussion.

Round 2

Reviewer 2 Report

General comment

It is a well-done work and the major issues that arose during the review have been addressed appropriately.

Minor comment

3.0 Results

3.5. Genome completeness and contamination analysis of identified prophages

Fig.5 and NOT Fig. 45
